# Ability of a Combined FIB4/miRNA181a Score to Predict Significant Liver Fibrosis in NAFLD Patients

**DOI:** 10.3390/biomedicines9121751

**Published:** 2021-11-24

**Authors:** Rodrigo Vieira Costa Lima, José Tadeu Stefano, Fernanda de Mello Malta, João Renato Rebello Pinho, Flair José Carrilho, Marco Arrese, Claudia P. Oliveira

**Affiliations:** 1Laboratório de Gastroenterologia Clínica e Experimental LIM-07, Division of Clinical Gastroenterology and Hepatology, Hospital das Clínicas HCFMUSP, Department of Gastroenterology, Faculdade de Medicina, Universidade de Sao Paulo, Sao Paulo 05403-000, SP, Brazil; rodrigovieiracostalima@gmail.com (R.V.C.L.); jose.tadeu@fm.usp.br (J.T.S.); femalta@yahoo.com (F.d.M.M.); jrrpinho@usp.br (J.R.R.P.); fjcarril@usp.br (F.J.C.); 2Departamento de Gastroenterología, Escuela de Medicina, Pontificia Universidad Católica de Chile, Santiago 833-0024, Chile; marrese@med.puc.cl; 3Centro de Envejecimiento y Regeneracion (CARE), Departamento de Biologia Celular y Molecular, Facultad de Ciencias Biologicas, Pontificia Universidad Católica de Chile, Santiago 833-0024, Chile

**Keywords:** non-alcoholic fatty liver disease, nonalcoholic steatohepatitis, microRNAs, miR-181a, liver fibrosis, FIB-4

## Abstract

Liver biopsy is the gold standard for assessing fibrosis, but there is a need to seek non-invasive biomarkers for this purpose. The aim of this study was to evaluate the correlation between the serum levels of the microRNAs miR-21, miR-29a, miR-122, miR-155 and miR-181a and the phenotypic expression of NAFLD. A cross-sectional study was carried out on 108 NAFLD patients diagnosed by liver biopsy. FIB-4 and NAFLD fibrosis scores were calculated. The comparison between the distributions of microRNA values according to the presence or absence of histological fibrosis (F2–F4) was performed. A multivariate logistic regression analysis was performed to build a score for predicting fibrosis using FIB-4 and Ln (miR-181a) as independent variables. Only miR-181a showed a statistical difference between patients with significant liver fibrosis (>F2) and those without (F0–F1) (*p* = 0.017). FIB-4 revealed an AUC on the ROC curve of 0.667 to predict clinically significant fibrosis (F2–F4). When assessed using the score in association with Ln (miR-181a), there was an improvement in the ROC curve, with an AUC of 0.71. miR-181a can be used as a non-invasive method of predicting fibrosis in NAFLD, and an association with FIB-4 has the potential to increase the accuracy of each method alone.

## 1. Introduction

Non-alcoholic fatty liver disease (NAFLD) is the most common cause of liver disease, and its incidence has increased dramatically over the past few years, following the rise in obesity worldwide [1,2,3]. An American study in an adult population of various ethnicities, which included more than 4000 participants, showed a high prevalence of steatosis (56.9%) and advanced fibrosis (5.5%) assessed by elastography [4]. Nonalcoholic steatohepatitis (NASH) is considered the progressive form and histological analysis is still the gold standard in the evaluation of fibrosis, which is the main prognostic factor for NAFLD, as well as hepatocellular ballooning and inflammation [5,6]. Although it has been established that numerous genetic and environmental factors influence the progression of NAFLD, no single diagnostic method has been able to replace liver biopsy in the identification of patients at higher risk of adverse liver outcomes. Nevertheless, due to the risk of complications, the high cost and the need of a specialized center, liver biopsy is not always performed. All these factors limit its use on a large scale, particularly in a disease as prevalent as NAFLD. Therefore, the search for non-invasive diagnostic and prognostic methods to select patients who are more likely to progress to more severe forms of the disease has recently increased [7,8,9].

Many biomarkers have already been studied to try to predict the presence of NASH in patients with NAFLD, as well as to assess the degrees of fibrosis in patients with NASH. Some non-invasive tests can be indirect fibrosis markers, such as the Fibrosis 4 index (FIB-4) and the NAFLD Fibrosis Score (NFS) [10]. Others can be direct markers, which estimate collagen deposition [11] and extracellular matrix turnover, such as PRO-C3 [12]. The biggest advantage of laboratory scores, such as NFS and FIB-4, is the low cost and accessibility. They can serve as a screening and selection of patients who would need specialized evaluation. Both NFS and FIB-4 are cost-effective, have high reproducibility and are sensitive to exclusion of patients with advanced fibrosis. They have been validated externally in different ethnicities with NAFLD, including in a Latin American population [13], and are recommended as tools for non-invasive fibrosis assessment by the latest consensus of the main societies. However, they have only moderate specificity in the identification of advanced fibrosis [2,14].

MicroRNAs consist of small RNAs of 18–25 nucleotides, which can regulate gene expression and protein translation. They can interfere in all aspects of cell activity, such as proliferation, differentiation, metabolism, apoptosis and carcinogenesis [15]. Because they are protected from degradation by RNases, in addition to being resistant to the extremes of temperature and pH, they can be a potential biomarker, both diagnostic and prognostic, for several diseases [16,17].

Based on the participation of microRNAs in oxidative stress, lipotoxicity, inflammatory cytokines, endothelial injury, adipocytokines and interaction with the intestinal microbiota, there was an interest in evaluating them as biomarkers in NAFLD. Among the most studied microRNAs in NAFLD are miR-21, miR-29a, miR-122, miR-155 and miR-181a [18,19,20,21,22,23,24,25,26,27,28,29]. Several factors can influence the levels of circulating miRNAs, such as age, sex, ethnicity and external interferences such as temperature and stress.

Recently, a consensus of international experts proposed that the name of the disease was changed from NAFLD to metabolic dysfunction-associated fatty liver disease (MAFLD) [9]. However, in this new clinical definition, the histological diagnosis of NASH was not included and other causes such as alcohol and viral hepatitis can be associated. In the present study, we chose to use NAFLD nomenclature instead of MAFLD due to the fact that our sample comprises only biopsy-proven NAFLD patients without other etiologies. Therefore, we assessed the performance of several microRNAs in predicting the presence of significant liver fibrosis in a sample of Brazilian patients.

## 2. Methods

### 2.1. Clinical Design and Patient’s Selection

A cross-sectional retrospective study was carried out on 108 biopsy-proven non-alcoholic fatty liver disease (NAFLD) patients from the Hepatology Outpatient Clinic at the Hospital das Clínicas da Faculdade de Medicina da Universidade de São Paulo (HC-FMUSP), São Paulo, Brazil. The material for the analysis of the microRNA expression was obtained from a database of DNA samples stored in the Laboratório de Gastroenterologia Clínica e Experimental (LIM-07) do Departamento de Gastroenterologia e Hepatologia do HC-FMUSP, from patients who were followed at the Hepatology Outpatient Clinic, who attended routine examinations and agreed—freely and spontaneously—that their DNA could be stored for future research. The study was approved by the Ethics Committee (2269.092) of the HC-FMUSP and it was conducted following the ethical guidelines of the Declaration of Helsinki. Individuals of both sexes between 18 and 75 years old were included, after excluding other causes of chronic liver disease, such as hepatosplenic schistosomiasis, viral hepatitis B and C, autoimmune hepatitis, primary biliary cholangitis, Wilson’s disease, α1-antitrypsin deficiency and hemochromatosis. Patients with alcohol intake greater than 30 g of ethanol per day for males and greater than 20 g per day for females were also excluded [2].

### 2.2. Clinical and Biochemical Analysis

Demographic data and blood samples were collected on the day of the liver biopsy or within a period of up to 3 months thereafter, including sex, age, weight and the presence of comorbidities, such as type 2 diabetes (T2DM), hypertension and dyslipidemia. Metabolic syndrome (MetS) was defined by the “Adult Treatment Panel III” (ATP III) criteria: hypertriglyceridemia >150 mg/dL; HDL cholesterol <40 mg/dL in men and <50 mg/dL in women; fasting blood glucose >110 mg/dL; waist circumference >102 cm in men and >88 cm in women; systolic blood pressure (BP) >130 mmHg; or diastolic BP >85 mmHg) [30]. HOMA-IR (“Homeostasis Model Assessment—Insulin Resistance”) was used to assess insulin resistance, defined by fasting blood glucose (mg/dL)/18 × fasting insulin (IU/mL)/22.5). Patients with HOMA-IR greater than or equal to 2.5 were considered to have insulin resistance [31].

General laboratory data, including complete blood count (CBC), lipidogram, fasting blood glucose, fasting insulin, prothrombin activity time, albumin, total bilirubin level and fractions, aspartate aminotransferase (AST), alanine aminotransferase (ALT), gamma-glutamyl transpeptidase (γ-GT), hepatitis B (HBV) and C (HCV) serological markers (HBsAg, anti-HBs, anti-HBc IgM and IgG, anti-HCV), liver autoantibodies (anti-nuclear antibody, anti–smooth muscle antibody, anti-LKM1, anti-mitochondria antibody), ceruloplasmin, α1-antitrypsin and iron profile (iron level, ferritin, transferrin saturation and total iron-binding capacity), were recorded.

### 2.3. Calculation of Noninvasive Fibrosis Scores

Two non-invasive fibrosis scores (FIB-4 and NFS) were calculated in order to assess liver fibrosis in those patients included in the study. NFS and FIB-4 scores were calculated using the original formulas [10,32]. NFS: −1.675 + 0.037 × age (years) + 0.094 × BMI (kg/m^2^) + 1.13 × diabetes (yes = 1, no = 0) + 0.99 × AST/ALT ratio—0.013 × platelet count (×10^9^/L)—0.66 × albumin (g/dL). FIB-4 score: age (years) × AST (U/L)/platelets (10^9^/L) × √ALT (U/L). 

### 2.4. MicroRNA Expression and Analysis

#### 2.4.1. Samples

Plasma or serum samples were stored at −80 °C and used for total RNA isolation. The total RNA was extracted using the TRIzol^®^ LS reagent (Ambion^®^ by Life Technologies, Carlsbad, CA, USA), which included a stage of organic extraction with phenol followed by a stage of precipitation of the RNA with alcohol. The manufacturer’s guidelines were followed. The RNA was eluted in 30 μL of elution buffer (Ambion^®^, Thermo Fisher Brand, Foster City, CA, USA) and stored at −80 °C until use.

#### 2.4.2. RNA Quantification

RNA purity was evaluated using the NanoDropTM 1000 Spectrophotometer (Fisher Scientific Technologies, Waltham, MA, USA). RNA samples with a 260/280 ratio between 1.7 and 2 were considered adequate. RNA quantification was assessed using the fluorometric method on the Qubit^®^ equipment (Invitrogen^TM^, Thermo Fisher Brand, Carlsbad, CA, USA).

#### 2.4.3. Reverse Transcription and cDNA Synthesis

After the quality analysis and subsequent quantification of the extracted RNA, a reverse transcription reaction was performed for the synthesis of complementary DNA (cDNA). For this purpose, the TaqMan^®^ MicroRNA Reverse Transcription Kit (Applied Biosystems, Thermo Fisher Brand, Foster City, CA, USA) was used. The cDNA was synthesized using specific primers for each microRNA studied. The reverse transcription reaction was performed with a final volume of 15 μL. The experiment was carried out using 75 ng of total RNA in the following steps: 16 °C for 30 min, then 42 °C for 5 min and a final step of 85 °C for 5 min for enzyme inactivation. 

#### 2.4.4. Detection of miRNAs by Real-Time PCR

The cDNA was used in the miRNA expression assays. Pre-designed miRNA expression assays (hsa-miR-21, hsa-miR-29a, hsa-miR-122, hsa-miR-155 and hsa-miR-181a) were used (Applied Biosystems, Thermo Fisher Brand, Foster City, CA, USA). The controls used in the reactions were ribosomal RNA RNU48 (Applied Biosystems, Thermo Fisher Brand, Foster City, CA, USA) and Spike-In C. elegans miR-39 (Qiagen^®^, Hilden, Germany). The reaction was performed using the TaqMan Universal PCR master mix II kit (Applied Biosystems, Thermo Fisher Brand, Foster City, CA, USA). Two microliters of the cDNA (5 ng/μL) were used, with a final reaction volume of 20 μL. Detection and analysis were performed on the 7500 Fast Real-Time PCR System (Applied Biosystems, Thermo Fisher Brand, Foster City, CA, USA), under the following cycling conditions: 50 °C for 20 s, 95 °C for 10 min (forty 15-s cycles) and 60 °C for 1 min. Each sample was analyzed in triplicate.

#### 2.4.5. Analysis of miRNAs Expression

The expression analysis was based on the value of the CT (cycle threshold) of each sample, which consists of the point from which fluorescence is detected. The CT is the starting point of the exponential amplification phase that is inversely correlated with the sample quantity. The greater the initial sample quantity, the lower the CT value. This means that the significant increase in fluorescence is detected earlier.

The CT values obtained for each sample were normalized using the CT values of the ribosomal RNA RNU48 (endogenous control) in order to minimize variations arising from the methodology itself [33]. The normalized CT value (ΔCT) of each sample was obtained as follows:ΔC_T_ = C_T(sample)_ − C_T(endogenous control)_

The expression of miRNAs was determined using the 2^−∆Ct^ algorithm [34].

### 2.5. Histological Analysis

The fragments of liver tissue obtained by a Tru-cut needle biopsy were evaluated in hematoxylin-eosin (HE), Masson’s trichrome and Perls stains. The liver samples were blindly evaluated by a specialized liver pathologist. The histological parameters were analyzed and quantified as follows: micro or macrovesicular steatosis and its zonal distribution, inflammatory infiltrate and its zonal distribution, portal and perivenular fibrosis and focal necrosis.

The inflammatory activity was measured using the NAFLD Activity Score (NAS), defined by the Pathology Committee of the NASH Clinical Research Network [35]. In order to categorize the degrees of inflammatory activity in two groups for statistical analysis, patients were divided into patients with significant inflammatory activity (NAS ≥ 4) and with an absence of significant inflammatory activity (NAS < 4). 

The degree of fibrosis was assessed by the staging proposed by Kleiner et al. [36]. For statistical analysis, the subdivision of grade 1 fibrosis was not considered, being all classified in the same category. Considering that the histological findings have a heterogeneous distribution in the samples, the degrees were assigned considering the cell and tissue densities of the variables. Thus, the gradation resulted both from the frequency with which variable appeared, as well as from the dimension of its expression. 

In order to categorize the degree of fibrosis in two groups for statistical analysis, patients were divided into patients with clinically significant fibrosis (degrees of fibrosis F2, F3 and F4) and with an absence of clinically significant fibrosis (degrees of fibrosis F0 and F1). This division was based on the recommendation of the European Association for the Study of the Liver (EASL), which considers pharmacological treatment for NASH only for patients with a fibrosis degree greater than or equal to 2. Therefore, the identification of this group of patients became clinically relevant [1].

### 2.6. Statistics

The data were presented according to the type of variable. For qualitative variables, frequencies and percentages were calculated. Values of the means, medians, standard deviations and interquartile ranges were calculated for the quantitative variables. The interquartile range is characterized as the difference between the third quartile (Q75%) and the first quartile (Q25%). 

The association between qualitative variables was assessed using Pearson’s chi-square test. When 25% or more of the expected values were less than 5, Fisher’s exact test was used. To verify the normality of the distribution of the microRNA values, the Kolmogorov–Smirnov normality test was used considering that the null hypothesis is a normal data distribution. Parametric tests were used in cases of non-rejection of the null hypothesis, while non-parametric tests were used in the case of rejection of the null hypothesis. 

The comparison between the distributions of the miRNA values in the fibrosis categories was performed using the Mann–Whitney U test, the student’s *t*-test and the Kruskal–Wallis test, according to the adherence of the data to the normal distribution. 

The correlation between quantitative variables was calculated using Spearman’s correlation coefficient. Graphs of the distribution of microRNAs according to qualitative variables were presented as boxplots. 

The level of significance adopted was 5% for all hypothesis tests. The analyzes were performed using the statistical software SPSS for Windows v.25.

A multivariate logistic regression analysis was performed to construct a score for the prediction of fibrosis using FIB-4 and Ln (miRNA-181a) as independent variables. ROC curves were constructed to verify the accuracy of the score (FIB4–miRNA181a) and of FIB-4.

Quantitative variables are presented as the mean ± SD or as the median and interval for non-parametric data. The variables were compared using the student’s *t* test with the SPSS v.20.0 software (IBM SPSS Statistics for Windows, Armonk, NY, USA). In addition, correlation analyses of miRNA expression and the biochemical parameters of the patients were performed using Spearman’s non-parametric correlation with the GraphPad PRISM software, version 5.01 (GraphPad Software Inc, La Jolla, CA, USA). In all tests, a value of *p* < 0.05 was considered statistically significant.

## 3. Results

The study sample consisted of 108 patients with biopsy-proven NAFLD. The patients included in the study are shown in Table 1, according to the degree of fibrosis in the liver biopsy. Clinically significant fibrosis (F2–F4) was present in 42.6% (46/108) of the cases. Among these patients, five had fibrosis grade F4 (4.6%), 22 grade F3 (20.4%) and 19 grade F2 (17.6%). Dyslipidemia was more prevalent in patients with an absence of clinically significant fibrosis (F0–F1: 86.7%) compared with 65.9% of those who had a clinically significant fibrosis (F2–F4) (*p*= 0.012). Additionally, the HDL cholesterol value showed statistically significant differences between the groups (*p* = 0.049), unlike the other cholesterol fractions. The ALT, AST and γ-GT evaluations also showed statistically significant differences between patients according to the degrees of fibrosis (*p* < 0.05), with higher levels in patients with significant fibrosis (F2–F4). The number of platelets was lower, with statistical significance, in patients with more advanced fibrosis (*p* = 0.034). Other characteristics did not show statistically significant differences between the groups analyzed. 

MicroRNA expression levels were assessed according to the degree of fibrosis. Given the non-normal distribution of the microRNAs, the Kruskal–Wallis test was performed. None of the microRNAs showed statistically significant differences between the degrees of fibrosis (*p* > 0.05) when each degree was evaluated separately. 

When we evaluated microRNAs expression in the groups considering the presence of clinically significant fibrosis (F2–F4) or the absence of clinically significant fibrosis (F0–F1), miRNA-181a showed statistically significant differences between the two groups (*p* = 0.017), with reduced expression in patients with clinically significant fibrosis (F2–F4). On the other hand, the values of miRNA-181a did not demonstrate statistical difference in other histological parameters such as hepatocellular ballooning, inflammation, steatosis or NAS activity. No other micro-RNA (miRNA-21, miRNA-29a, miRNA-122 and miRNA-155) demonstrated accuracy in detecting significant fibrosis at a comparable level to miRNA-181a (Table 2), and they did not show statistical differences in terms of other histological parameters.

When analyzing the values of microRNA expression according to the FIB-4 and NFS categorizations, no statistically significant differences were found between the groups (*p* > 0.05) (Table 3 and Table 4). However, in our population, neither FIB-4 nor NFS were good predictors for the exclusion of clinically significant fibrosis (F2–F4), considering histological analysis as the gold standard, when used alone, with an area under the curve (AUC) of 0.616 for the NFS and 0.698 for the FIB-4 (Figure 1). 

In order to improve the accuracy in detecting more severe stages of fibrosis, we added miR-181a to FIB-4 as a predictor of clinically significant fibrosis (F2–F4). A logistic regression analysis was performed, as shown in Table 5.

Based on the coefficients of the multivariate model, the following score was established:Score (FIB4–miRNA181a) = −3.641 + 1.334 * FIB-4 − 0.269 * ln(miR-181)

The score value was calculated for all sample participants and then an ROC curve was drawn, where the score obtained an AUC of 0.751, higher than that of the FIB-4 alone, which was 0.698. (Figure 2).

In order to compare the characteristics of the patients who had greater or lesser inflammatory activity in the biopsy assessed by the NAS, we divided our population into two groups: patients with NAS < 4 and patients with NAS ≥ 4. There was no statistically significant difference in the analysis of expression of the microRNAs studied between the two groups.

To assess the correlation between the microRNAs and the NAS histological score, Spearman’s correlation coefficients were calculated (Table 6). No correlation was found between microRNAs and NAS values in our sample.

## 4. Discussion

Histological analysis is still the gold standard in the assessment of liver fibrosis, a key prognostic factor for NAFLD. However, liver biopsy has a number of limitations that had favored the continuous search and development of alternative non-invasive strategies to detect advanced liver fibrosis. The use of microRNAs, as serum biomarkers of both inflammation and fibrosis, has gained attention in recent years. In the present study, we evaluated the serum levels of five circulating microRNAs in a well-characterized group of biopsy-proven NAFLD, and found an unprecedented decrease in miR-181a in the blood of patients with clinically significant fibrosis (F2–F4) when compared with those with an absence of clinically significant fibrosis (F0–F1). Furthermore, we created a score using miR-181a associated with FIB-4 that showed an increased ability to identify patients with clinically significant fibrosis (F2–F4). 

There are few studies that have tried to correlate the histological aspects of NAFLD with the expression of microRNAs, and particularly that differentiate significant fibrosis (F2–F4) from mild fibrosis or an absence of fibrosis (F1–F0). 

MiR-181a is abundantly expressed in several tissues, including the liver [37]. There are reports of the role of miR-181a in fibrogenesis in other organs, such as the lung [38], favoring that it may play a role in liver fibrosis. Although recent studies have shown that miR-181a expression is higher in the serum of NAFLD patients than in healthy controls [37,39], none of them assessed the differences in the serum level of mir-181a according to the degrees of fibrosis demonstrated by histological analysis, as our study demonstrated.

There is another Brazilian study in cirrhotic patients that did not demonstrate statistical difference in the serum level of miR-181a between patients with cirrhosis and healthy controls. However, differently to our study, the authors did not evaluate the serum levels in different degrees of fibrosis [21]. On the other hand, Gupta et al. showed increased expression of miR-181a in the tissue of cirrhotic livers, possibly associated with increased fibrogenesis via TGF-β [40]. However, these data do not contradict our findings since we measured miR-181a in the blood and not in liver tissue.

One important differential of our study is the construction of a new score, associating miR-181a with FIB-4, which increased the ability to identify patients with clinically significant fibrosis (F2–F4). Some scores were developed with the objective of selecting the patients with the highest risk of advanced fibrosis. Those who are already better established and have been externally validated are included in the Fibrosis-4 (FIB-4) and the NAFLD fibrosis score (NFS). However, only a few scores such as HEPAMET [41] have comparable accuracy to our score in predicting significant fibrosis (including also F2, beyond F3 and F4). Recently, a Latin American group validated all three scores (HEPAMET, FIB-4 and NFS) in a Latin America population, including Brazilians, also demonstrating good accuracy in terms of predicting significant fibrosis [14]. A meta-analysis also demonstrated the accuracy of these biomarkers in detecting advanced fibrosis, showing an area under the curve (AUC) of 0.84 [42].

Another advantage of our study beyond the association of miR181a plus FIB-4 is that single-center studies reduce interobserver variation in the interpretation of liver biopsy findings. Another advantage is the fact that we could study specific aspects of the Brazilian population, an admixed population, as ethnicity could influence both NAFLD aspects and the serum level of microRNAs. Recently, we studied PNPLA3 in a Brazilian population, and although it was an admixed population, PNPLA3 predicted NAFLD in our population similarly as validated in other ethnicities [43]. However, in epigenetics, this is the first Brazilian study in patients with NAFLD confirmed by biopsy.

In daily practice, some studies have shown the inferiority of NFS and FIB-4 in the evaluation of fibrosis when compared to other methods, such as FibrometerV2G and FibroScan^®^ [44]. A cross-sectional Brazilian study also showed the superiority of FibroScan^®^ over scores such as the NFS and FIB-4 for the diagnosis and exclusion of advanced fibrosis in patients with NAFLD [45].

On the other hand, despite being a good marker in the assessment of fibrosis, FibroScan^®^ had its accuracy clearly reduced in patients with BMI above 28 [46], which corresponds to a significant percentage of patients with NAFLD. In our sample, 93.7% of the patients were overweight (26%) or obese (67.7%).

Due to its complex pathogenesis, it is difficult to find a method or biomarker that alone can predict fibrosis in NAFLD [7]. This encourages us to incorporate new serum parameters and develop new scores to improve non-invasive fibrosis laboratory evaluation, with microRNAs being interesting options in this regard. In the present study, we developed a new score using miR-181a associated with FIB-4 that showed an increased ability to identify patients with clinically significant fibrosis (F2–F4).

Regarding other microRNAs in our study, the analysis of the expression of miR-21 in serum did not correlate with any of the clinical, laboratory or histological variables, including the degree of fibrosis in our population. There was also no correlation with the miR-122 level and the degrees of fibrosis of the patients. Yamada et al. found an increased serum level of miR-21 in patients with NAFLD compared to healthy controls [47]. However, another study showed exactly the opposite [48], using similar methodology, but in different populations, reinforcing ethnic issues and environmental involvement at the serum level and, consequently, in the clinical use of microRNAs.

Although some studies suggested that miR-29 would have reduced serum levels in patients with advanced fibrosis [39], our study was unable to confirm this finding. Blaya et al. showed a higher expression of miR-155 in cirrhotic patients. However, they were unable to demonstrate statistically significant differences between patients with compensated or decompensated cirrhosis [29]. These data are consistent with our results, where no differences in serum miR-155 levels were demonstrated between the studied disease phenotypes.

It is important to highlight some limitations and possible biases in our study. The average age of the patients evaluated (56.8 years) does not necessarily reflect the behavior of NAFLD in younger patients. There was a clear predominance of female patients (78.7%). Serum microRNA levels were not measured in healthy controls for comparative purposes. Another point to be considered is that the degree of fibrosis by biopsies was considered the gold standard, but since all of them were performed percutaneously, we must admit the possibility of sample error, given the non-uniform distribution of the disease in the liver parenchyma. Furthermore, since the biopsy is an invasive procedure, few patients in our series had fibrosis of grade F4 (5/108), because those with more advanced cirrhosis were probably not candidates for biopsy. In addition, it was a single-center study carried out in a tertiary hospital, with possible patient selection bias, and the small sample size cannot guarantee that these results can be extrapolated to the general public.

Despite the limitations, such as the small number of subjects, this study will be useful for future research on the establishment of a non-invasive approach to NAFLD, as it characterized the role of some microRNAs, correlating their expression with the different phenotypes of the disease, mainly its correlation with findings on liver biopsy.

## 5. Conclusions

In conclusion, the incorporation of FIB-4 in the construction of the FIB4–miRNA181a Score improved the accuracy in the identification of clinically significant fibrosis, with superior results in relation to the use of each method alone. Furthermore, future research can evaluate the usefulness of miR-181a in terms of the accuracy of fibrosis assessment when used in combination with other methods, such as TE, serological scores or other biomarkers.

## Figures and Tables

**Figure 1 biomedicines-09-01751-f001:**
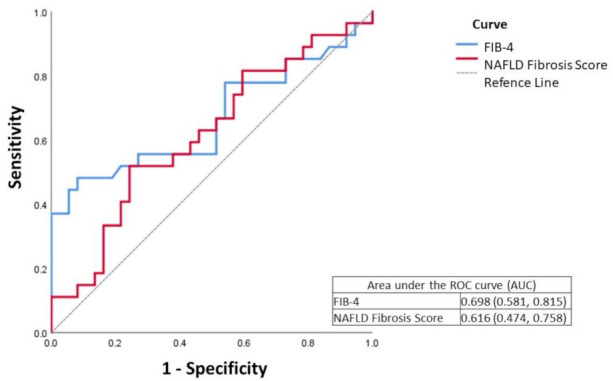
ROC curve of FIB-4 (Fibrosis index-4) and NFS (NAFLD fibrosis score) in fibrosis assessment.

**Figure 2 biomedicines-09-01751-f002:**
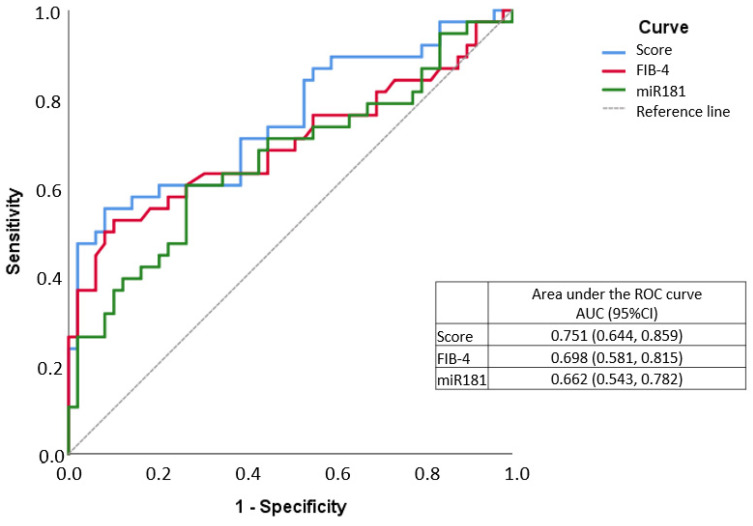
ROC curve of Fibrosis index-4 (FIB-4)/miR-181a score in fibrosis assessment.

**Table 1 biomedicines-09-01751-t001:** Characteristics of patients according to the degree of fibrosis.

Characteristic	Total	Fibrosis		*p* Value
		F0/F1	F2–F4	
	n (%)	n (%)	n (%)	
	n = 108	n = 62	n = 46	
Sex				0.567 ^1^
Male	23 (21.3%)	12 (19.4%)	11 (23.9%)	
Female	85 (78.7%)	50 (80.6%)	35 (76.1%)	
Age (years)				0.634 ^3^
Mean (SD)	56.8 ± 9.4	56.8 ± 8.2	56.7 ± 10.9	
Med (min-max)	58 (27–74)	58 (33–69)	60.5 (27–74)	
Type 2 diabetes	70 (67.3%)	36 (60.0%)	34 (77.3%)	0.064 ^1^
Dyslipidemia	81 (77.9%)	52 (86.7%)	29 (65.9%)	0.012 ^1^
Hypertension	68 (66.0%)	39 (66.1%)	29 (65.9%)	0.984 ^1^
BMI (kg/m^2^)				
Mean (SD)	32.2 ± 5.8	33.2 ± 6.1	31.0 ± 5.1	0.090 ^4^
Normal	6 (6.3%)	3 (5.3%)	3 (7.7%)	0.300 ^2^
Overweight	25 (26.0%)	12 (21.1%)	13 (33.3%)	
Obese	65 (67.7%)	42 (73.7%)	23 (59.0%)	
Fasting blood glucose				0.053 ^3^
Mean (SD)	118.9 ± 40.3	114.7 ± 42.4	124.7 ± 37.0	
Med (min-max)	105.5 (73–273)	100 (73–273)	109 (79–220)	
Insulin				0.548 ^3^
Mean (SD)	21.4 ± 14.2	21.5 ± 15.7	21.2 ± 11.9	
Med (min-max)	18.3 (3.7–70.5)	17.4 (5.7–70.5)	19.8 (3.7–63.4)	
Insulin resistance index (HOMA)				0.182 ^3^
Mean (SD)	6.4 ± 5.0	6.3 ± 5.6	6.5 ± 4.1	
Med (min-max)	4.9 (0.8–23)	4.4 (1.2–23)	5.7 (0.8–17.3)	
Metabolic syndrome	83 (80.6%)	47 (78.3%)	36 (83.7%)	0.495 ^1^
Aspartate aminotransferase				<0.001 ^3^
Mean (SD)	40.7 ± 37.6	28.4 ± 18.3	57.1 ± 49.1	
Med (min-max)	32 (10–248)	23 (10–141)	41 (13–248)	
Alanine aminotransferase				<0.001 ^3^
Mean (SD)	52.2 ± 51.9	37.5 ± 23.5	71.8 ± 70.4	
Med (min-max)	38 (13–479)	29 (13–149)	52.5 (19–479)	
Gamma-glutamyl transferase				0.044 ^3^
Mean (SD)	91.0 ± 99.6	68.9 ± 65.4	120.8 ± 127.2	
Med (min-max)	54 (12–476)	45 (12–389)	67.5 (13–476)	
Total cholesterol				0.729 ^4^
Mean (SD)	194.4 ± 46.5	193 ± 45.1	196.2 ± 48.7	
Med (min-max)	189 (86–313)	193 (86–293)	189 (95–313)	
HDL cholesterol				0.049 ^3^
Mean (SD)	47.0 ± 12.9	49 ± 13.4	44.3 ± 11.8	
Med (min-max)	45 (24–100)	48 (25–100)	42.5 (24–75)	
LDL cholesterol				0.370 ^4^
Mean (SD)	115.7 ± 41.0	112.5 ± 36.4	120 ± 46.6	
Med (min-max)	114 (22–245)	114 (32–207)	112 (22–245)	
Triglycerides				0.219 ^3^
Mean (SD)	162.8 ± 68.7	154.8 ± 64.6	173.6 ± 73.4	
Med (min-max)	151 (50–433)	141 (50–319)	156.5 (74–433)	
Albumin				0.753 ^4^
Mean (SD)	4.63 ± 0.3	4.6 ± 0.3	4.6 ± 0.3	
Med (min-max)	4.6 (3.9–5.2)	4.7 (3.9–5.2)	4.6 (4–5.1)	
Platelets				0.034 ^3^
Mean (SD)	239.4 ± 69.1	254.3 ± 61.8	220.4 ± 73.9	
Med (min-max)	*245* (*92–484*)	*248* (*146–484*)	*218.5* (*92–385*)	

BMI: body mass index; SD: standard deviation; med: median; min: minimum value; max: maximum value. ^1^ Pearson’s chi-square test; ^2^ Fisher’s exact test; ^3^ Mann–Whitney test; ^4^ Student’s t test.

**Table 2 biomedicines-09-01751-t002:** Expression of microRNAs according to absence of clinically significant fibrosis (F0–F1) or the presence of clinically significant fibrosis (F2–F4).

Characteristic	Total	Fibrosis		*p* ^b^
	(n = 108)	F0/F1 (n = 62)	F2–F4 (n = 46)	
miRNA-21	0.14 (0–37.98)	0.14 (0–37.98)	0.12 (0–16.34)	0.2033
miRNA-29a	0.02 (0–3.55)	0.02 (0–3.55)	0.03 (0–3.18)	0.7513
miRNA-122	0.02 (0–5.72)	0.02 (0–0.37)	0.02 (0–5.72)	0.5133
miRNA-155	0.004 (0–8.74)	0.004 (0–0.40)	0.003 (0–8.74)	0.9543
miRNA-181a	0.003 (0–1.07)	0.004 (0–0.39)	0.002 (0–1.07)	0.0173

Data expressed as median (minimum-maximum). ^b^: Mann–Whitney test.

**Table 3 biomedicines-09-01751-t003:** Comparison between microRNAs and Fibrosis index-4 (FIB-4) categorization.

miRNA	n	FIB-4 Categorization	Median (IQR)	*p* Value ^1^
miRNA-21	54	<1.3 (absence of significant fibrosis)	0.143 (0.427–0.015)	0.778
	26	1.3–2.67 (indeterminate)	0.131 (0.417–0.027)	
	9	>2.67 (presence of advanced fibrosis)	0.063 (0.927–0.006)	
miRNA-29a	54	<1.3 (absence of significant fibrosis)	0.025 (0.061–0.008)	0.602
	26	1.3–2.67 (indeterminate)	0.017 (0.094–0.006)	
	9	>2.67 (presence of advanced fibrosis)	0.008 (0.071–0.004)	
miRNA-122	53	<1.3 (absence of significant fibrosis)	0.017 (0.047–0.005)	0.688
	26	1.3–2.67 (indeterminate)	0.011 (0.046–0.002)	
	9	>2.67 (presence of advanced fibrosis)	0.014 (0.058–0.004)	
miRNA-155	53	<1.3 (absence of significant fibrosis)	0.004 (0.012–0.001)	0.630
	26	1.3–2.67 (indeterminate)	0.003 (0.007–0.001)	
	9	>2.67 (presence of advanced fibrosis)	0.002 (0.010–0.0003)	
miRNA-181a	52	<1.3 (absence of significant fibrosis)	0.004 (0.014–0.001)	0.277
	26	1.3–2.67 (indeterminate)	0.002 (0.009–0.001)	
	9	>2.67 (presence of advanced fibrosis)	0.001 (0.019–0.0003)	

IQR: Interquartile range (3rd Quartile–1st Quartile). ^1^ Kruskal–Wallis test.

**Table 4 biomedicines-09-01751-t004:** Comparison between microRNAs and the categorization of the NAFLD fibrosis score (NFS).

miRNA	n	NFS Categorization	Median (IQR)	*p* Value ^1^
miRNA-21	24	<−1.45 (absence of significant fibrosis)	0.164 (0.685–0.119)	0.603
	38	−1.45–0.675 (indeterminate)	0.137 (0.512–0.051)	
	2	>0.675 (presence of advanced fibrosis)	0.179 (NA–0.001)	
miRNA-29a	24	<−1.45 (absence of significant fibrosis)	0.030 (0.055–0.012)	0.987
	38	−1.45–0.675 (indeterminate)	0.026 (0.098–0.007)	
	2	>0.675 (presence of advanced fibrosis)	0.038 (NA–0.007)	
miRNA-122	24	<−1.45 (absence of significant fibrosis)	0.018 (0.044–0.010)	0.999
	37	−1.45–0.675 (indeterminate)	0.023 (0.048–0.005)	
	2	>0.675 (presence of advanced fibrosis)	0.038 (NA–0.003)	
miRNA-155	34	<−1.45 (absence of significant fibrosis)	0.003 (0.010–0.001)	0.518
	37	−1.45–0.675 (indeterminate)	0.005 (0.013–0.001)	
	2	>0.675 (presence of advanced fibrosis)	0.007 (NA–0.002)	
miRNA-181a	24	<−1.45 (absence of significant fibrosis)	0.004 (0.012–0.002)	0.865
	37	−1.45–0.675 (indeterminate)	0.003 (0.009–0.001)	
	2	>0.675 (presence of advanced fibrosis)	0.014 (NA–0.0003)	

IQR: Interquartile range (3rd Quartile–1st Quartile). ^1^ Kruskal–Wallis test.

**Table 5 biomedicines-09-01751-t005:** Multivariate logistic regression model with clinically significant fibrosis (F2–F4) and predictors FIB-4 and Natural Logarithm of miR-181a expression.

Equation Variables	β	S.E.	OR (IC95%)	*p*
FIB-4	1.334	0.433	3.8 (1.63–8.87)	<0.01
Ln(miR-181)	−0.269	0.138	1.31 (1–1.72)	0.05
Constant	−3.641	1.013		

β: coefficient of the regression model; SE: standard error of the β coefficient; OR: odds ratio; CI: confidence interval; Ln: natural logarithm; FIB-4: Fibrosis 4 index.

**Table 6 biomedicines-09-01751-t006:** Correlation between microRNAs and NAFLD activity score (NAS).

Correlation (*r*) ^1^	miRNA-21	miRNA-29	mirRNA-122	miRNA-155	miRNA-181
*p*^2^ Value					
NAFLD activity score (NAS)	−0.048	0.007	0.061	−0.005	−0.075
	0.622	0.942	0.533	0.958	0.449

NAFLD: Non-alcoholic fatty liver disease. ^1^ Spearman’s correlation coefficient; ^2^ under null hypothesis that there is no correlation between the variables (H0: ρ = 0).

## Data Availability

The raw data required to reproduce these results are available from the corresponding author upon reasonable request.

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
