# Peer review of "Ability of a Combined FIB4/miRNA181a Score to Predict Significant Liver Fibrosis in NAFLD Patients"

_biomedicines, 2021, doi:10.3390/biomedicines9121751_

Round 1
Reviewer 1 Report
The manuscript by Vieira Costa Lima et al. is on the association of FIB-4 and miR-181a as a new score in order to detect clinically significant fibrosis in NAFLD.
The purpose of developing noninvasive biomarkers of fibrosis is a real need at present and therefore the manuscript is of interest and original even if presents some limitations that the Authors are aware of and list in the discussion.
Minor comments.
It requires just minor English revision.
Author Response
SUBJECT: Response to reviewer comments
Manuscript No.: biomedicines-1432781
Title: entitled: "Ability of a combined FIB4/miRNA181a score to predict significant liver fibrosis in NAFLD patients"
Thank you for considering our manuscript for publication in Biomedicines. We are appreciative of the reviewers’ comments. After carefully reviewing the comments, we have revised the manuscript to address each of the reviewers’ concerns. Our point-by-point answers are provided below. We believe that we have adequately addressed all the issues raised by the reviewers. All changes in the manuscript have been highlighted in yellow. We expect that the revised manuscript will be now suitable for publication in Biomedicines.
Response to Reviewer #1’s comments
The manuscript by Vieira Costa Lima et al. is on the association of FIB-4 and miR-181a as a new score in order to detect clinically significant fibrosis in NAFLD.
The purpose of developing noninvasive biomarkers of fibrosis is a real need at present and therefore the manuscript is of interest and original even if presents some limitations that the Authors are aware of and list in the discussion.
Minor comments.
It requires just minor English revision.
Comments 1
Thank you. As suggested, the manuscript will be revised and edited by a native English speaker at American Journal Experts, a division of Research Square (Durham, North Carolina, USA), however, we need more time to do this after the deadline (today).

Reviewer 2 Report
1-It is a very welcomes study.
2-However, it is a very short study sample raising concerns while generalizing these findings to general public. This limitation should be addressed properly.
3-Comaprison with the normal patient group should be done to make any conclusion about this possible association.
Author Response
Response to Reviewer #2’s comments
- It is a very welcomes study.
Comments 1
Thank you for considering our manuscript for publication in Biomedicines
- However, it is a very short study sample raising concerns while generalizing these findings to general public. This limitation should be addressed properly.
Comments 2
The small sample size certainly limits the extrapolation of our results to the general public. We agree with this limitation of our study and we emphasize it in the text as suggested.
- Comparison with the normal patient group should be done to make any conclusion about this possible association.
Comments 3
Studying the expression of microRNAs in the general population could help us to understand their biological behavior. However, all patients in our sample consisted of patients with NAFLD and our objective was not to diagnose this condition, but rather to stratify fibrosis in those with the disease.

Reviewer 3 Report
In the present manuscript the Authors report the results of a cross-sectional study evaluating the ability of FIB-4 in association with several serum measured miRNAs to predict the presence of significant liver fibrosis (>=F2) in a sample of patients with NAFLD from Brazil. The topic is certainly of interest because, as the Authors point out, there is an urgent need to replace liver biopsy with a well-performing non-invasive technique to identify liver fibrosis. The manuscript is well written and clear. Still, I have some reservations.
- When new biomarkers are introduced, the best methodological way to proceed is to divide the cohort into a derivation and validation groups, to show whether the model is robust. This was not performed in the current study, maybe because of the small sample size and in my opinion it reduces the validity of the results.
- FIB-4 and NFS were originally introduced to identify advanced fibrosis (F3-F4) rather than F2. This might account for the low AUC identified in the present study.
- In table 1, you report the word “media” for continuous variables. Is this a mean or median?
- Please at least report in the text the number and % of patients with F3 and F4 in your cohort.
- I would report the AUC as a value and corresponding 95% confidence intervals.
- Apart from AUC analysis, it could be interesting to report a youden index cut-off and evaluate the performance of the score in terms of sensitivity and specificity. The AUC is not that good and this is a major limitation of the current study. How would implementation of this score work in clinical practice to re-classify patients in the correct category as compared with FIB-4 or NFS?
7. In the introduction section, when discussing the epidemiology of NAFLD, I would mention more recent studies performed with transient elastography in the general population reporting not just the prevalence of steatosis, but also that of advanced liver fibrosis (e.g. doi: 10.1097/HJH.0000000000002835)
Author Response
Response to Reviewer #3’s comments
In the present manuscript the Authors report the results of a cross-sectional study evaluating the ability of FIB-4 in association with several serum measured miRNAs to predict the presence of significant liver fibrosis (>=F2) in a sample of patients with NAFLD from Brazil. The topic is certainly of interest because, as the Authors point out, There is an urgent need to replace liver biopsy with a well-performing non-invasive technique to identify liver fibrosis. The manuscript is well written and clear. Still, I have some reservations.
- When new biomarkers are introduced, the best methodological way to proceed is to divide the cohort into a derivation and validation groups, to show whether the model is robust. This was not performed in the current study, maybe because of the small sample size and in my opinion, it reduces the validity of the results
Comments 1
We fully agree with the idea that the best strategy would be to divide the cohort into validation and derivation groups. However, as our sample only included patients with liver biopsy, the number of patients was limited, which made this validation difficult. estimate of optimism we performed a Boostrap analysis (not shown in the results). Due to the importance of our findings, we think it is relevant that the data be published, allowing other researchers to carry out the validation in other populations.
FIB-4 and NFS were originally introduced to identify advanced fibrosis (F3-F4) rather than F2. This might account for the low AUC identified in the present study.
Comments 2
We understand and agree with this fact and believe that this certainly leads to a limitation of our data. The objective of using the F2 cutoff point was an attempt to select patients with clinically significant fibrosis, thus being the group that would benefit most from therapeutic interventions.
- In table 1, you report the word “media” for continuous variables. Is this a mean or median?
Comments 3
There was a typo and where it was written "media" should be written mean. Corrections have been made.
- Please at least report in the text the number and % of patients with F3 and F4 in your cohort.
Comments 4
There are 27/108 F3-F4 patients (25% of the total). 5 F4 patients (4.6% of the total) and 22 F3 patients (20.4% of the total). Total of 19/108 F2 patients (17.6%).
These data have been added to the text as suggested.
I would report the AUC as a value and corresponding 95% confidence intervals.
Comments 5
The 'areas under the curve' confidence interval values have been added to the figures 1 and 2 as suggested.
Apart from AUC analysis, it could be interesting to report a youden index cut-off and evaluate the performance of the score in terms of sensitivity and specificity. The AUC is not that good and this is a major limitation of the current study. How would implementation of this score work in clinical practice to re-classify patients in the correct category as compared with FIB-4 or NFS?
Comments 6
Below are the evaluations of the variables using the Youden score as suggested. Despite low sensitivity and specificity values, the score showed better performance than the other variables alone. We could establish a different cutoff to improve the sensitivity of the score and make it more attractive as a screening test in clinical practice, but we need more time to do this type of analyses.
- FIB 4
Area under the ROC curve (AUC): 0.698 (0.581, 0.815)
CRITERION: Youden
Number of optimal cutoffs: 1
Estimate
cutoff 1.5300000
Se 0.5128205
Sp 0.9000000
PPV 0.8000000
NPV 0.7031250
DLR.Positive 5.1282051
DLR.Negative 0.5413105
FP 5.0000000
FN 19.0000000
Optimal criterion 0.4128205
miR181
Area under the ROC curve (AUC): 0.662 (0.543, 0.782) CRITERION: YoudenNumber of optimal cutoffs: 1 Estimatecutoff 0.0019500Se 0.6052632Sp 0.7346939PPV 0.6388889NPV 0.7058824DLR.Positive 2.2813765DLR.Negative 0.5372807FP 13.0000000FN 15.0000000Optimal criterion 0.3399570
SCORE Area under the ROC curve (AUC): 0.751 (0.644, 0.859) CRITERION: YoudenNumber of optimal cutoffs: 1 Estimatecutoff 3.8000000Se 0.5526316Sp 0.9183673PPV 0.8400000NPV 0.7258065DLR.Positive 6.7697368DLR.Negative 0.4871345FP 4.0000000FN 17.0000000Optimal criterion 0.4709989
- In the introduction section, when discussing the epidemiology of NAFLD, I would mention more recent studies performed with transient elastography in the general population reporting not just the prevalence of steatosis, but also that of advanced liver fibrosis (doi: 10.1097/HJH.0000000000002835).
Comments 7
Data from this study were included in the introduction.

Round 2
Reviewer 3 Report
I have no further comments.